# Filtered Semantic Search via Vector Arithmetic

## Abstract

How can we retrieve search results that are both semantically relevant and satisfy certain filter criteria? Modern day semantic search engines are increasingly reliant on vector-based search, yet the ability to restrict vector search to a fixed set of filter criteria remains an interesting problem with no known satisfactory solution. In this note, we leverage the rich emergent structure of vector embeddings of pre-trained search transformers to offer a simple solution. Our method involves learning, for each filter, a vector direction in the space of vector embeddings, and adding it to the query vector at run-time to perform a search constrained by that filter criteria. Our technique is broadly applicable to any finite set of semantically meaningful filters, compute-efficient in that it does not require modifying or rebuilding an existing $k$-NN index over document vector embeddings, lightweight in that it adds negligible latency, and widely compatible in that it can be utilized with any transformer model and $k$-NN algorithm. We also establish, subject to mild assumptions, an upper bound on the probability that our method errantly retrieves irrelevant results, and reveal new empirical insights about the geometry of transformer embeddings. In experiments, we find that our method, on average, yields more than a 21% boost over the baseline (measured in terms of nDCG@10) across three different transformer models and datasets.

## 1    Introduction

Search is an age-old problem in the field of information retrieval. It has been revolutionized by modern machine learning systems, particularly by pre-trained transformer models Nogueira & Cho (2020); Devlin et al. (2018). When these models are fine-tuned for search—hereafter referred to as search transformers—they demonstrate state-of-the-art performance across a wide variety of information retrieval tasks, domains, and data modalities Yates et al. (2021). The natural language understanding capabilities of these search transformers have enabled users to pose queries in natural language and retrieve semantically relevant results.

A query for *shoes* in an e-commerce corpus retrieves not only product articles about *shoes* but also those on *boots, loafers, sneakers*, etc. In many situations, this is the intended behavior. However, semantic relevance is not the sole criterion for obtaining relevant results. Often, users seek results that are semantically relevant *and* meet specific keyword criteria. For example, a user might want to find *shoes* from a specific brand, such as *Nike*. In this case, the ideal results would be semantically relevant to *shoes*, such as *shoe* and *sneakers*, while exactly matching the brand *Nike*. In this paper, we study how one can retrieve results that simultaneously exhibit semantic relevance while satisfying certain filter criteria.

The fundamental principle of search transformers is to generate a vector embedding for every item in the corpus, henceforth referred to as documents, in a way that ensures that vector embeddings of relevant documents are positioned closer to each other in the vector space and vice versa. The resulting collection of document vectors forms the universe over which searches are conducted. At the time of a query, the query $q$ is converted into a vector, and the nearest neighbors of the query vector $\vec{v}_q$ are retrieved as the relevant results. The nearest neighbor search is executed either using an exact search or, more commonly, an approximate $k$-Nearest Neighbors ($k$-NN) method such as Hierarchical Navigable Small Worlds (HNSW) Malkov & Yashunin (2020) or Product Quantization (PQ) Jégou et al. (2011). Both approximate methods involve a computationally intensive step of building an index, which, while substantially increasing the initial setup time, significantly reduces downstream latency during query retrieval.

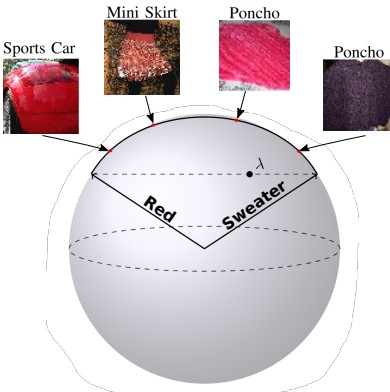

Figure 1: The "Red ↔ Sweater" continuum using `clip` for Tiny Imagenet. For an appropriate value of $\lambda$, the nearest neighbor of $\vec{v}_{Sweater} + \lambda\vec{\nu}_{Red}$, retrieves a relevant result *a red Poncho*.

Our method relies on the intuition that there exist emergent directions in the space of vector embeddings that correspond to different filters such as *Nike* or *Sony*, and that these directions can be found using a simple algorithm that employs linear probes with a limited amount of un-supervised data. At run-time, a query $q$ with a filter $f$ is vectorized into a simple linear combination, such as $\vec{v}_{q+f} + \lambda\vec{\nu}_f$, before being submitted to $k$-NN search. Here $\lambda$ is a scalar, $\vec{v}_{q+f}$ refers to the model embedding of the concatenated query and filter string, and $\vec{\nu}_f$ corresponds to the learned filter direction. Experimentally, this simple protocol achieves an improvement over the baseline of almost 21% across different datasets and models, as measured in terms of nDCG@10 and Recall@10.

We discover that other linear combinations also yield comparable performance. For illustration and theoretical analysis, we work with $\vec{v}_q + \lambda\vec{\nu}_f$, while for our experiments, we use $\vec{v}_{q+f} + \lambda\vec{\nu}_f$. The former greatly simplifies the illustration and theoretical analysis, while the latter allows for a more intuitive comparison with the experimental baseline. As shown in Figure 1, our method leverages the continuum of semantically meaningful linear combinations $\vec{v}_q + \lambda\vec{\nu}_f$ within transformer embeddings as we vary $\lambda$. Specifically, using image embeddings from `clip` with Tiny Imagenet Le & Yang (2015), we find that increasing $\lambda$ spans concepts from an image of a sweater (*Poncho*) to that of a red object (*red sports car*). For an appropriate value of $\lambda$, our method retrieves the relevant results. Thus, $\lambda$ serves as a hyperparameter in our approach, which we fix for a given dataset and model.

We also derive some theoretical results about our method and establish bounds on the probability of retrieving irrelevant results subject to a particular generative model with mild assumptions. We motivate these underlying assumptions with new insights into the geometry of search transformers, specifically regarding the isotropy of their output representations. Contrary to previous findings, our research indicates that search transformers do not exhibit the kind of anisotropy often assumed to indicate degenerate representations. For example, our findings suggest that for most search transformers, vector embeddings of random, unrelated documents are nearly orthogonal to one another, i.e., are not anisotropic, and they do not possess "rogue" dimensions that could disproportionately influence the nearest neighbor distance calculation.

Query latency is a critical metric in modern search applications; thus, the proposed method should not be computationally heavy and must not add considerable latency during runtime. Additionally, a solution that necessitates rebuilding of the $k$-nearest neighbors index is impractical due to the vast size of contemporary search corpora. Our method is computationally efficient, adds negligible latency, and does not require the index to be rebuilt. It can be seamlessly integrated with any existing search transformer without the need for fine-tuning. However, since our approach relies on the presence of emergent directions leverageable by search transformers, it is effective only for filters with semantic relevance. Concretely, the filters cannot be arbitrary alphanumeric strings, such as *serial number*, *size* or *price*. To summarize, our contributions include

1. A simple, lightweight, and low latency algorithm for semantic search with filters, independent of both the transformer model and the underlying $k$-NN search algorithm.

2. Experiments demonstrating an improvement of 21% in nDCG@10 compared to a baseline across three different transformer models and datasets.

3. A theoretical analysis of the approach, establishing accuracy bounds under a certain generative model, and empirical insights into the geometry of vector embeddings.

## 2 BACKGROUND AND RELATED WORK

One of the oldest and most widely used techniques for information retrieval is BM25 Robertson & Zaragoza (2009). It is based on keyword matching and frequency statistics, providing a robust, generalizable search solution for many use cases. However, keyword-based search lacks the semantic understanding of a pre-trained search transformer, and thus it cannot be used to retrieve semantically relevant results with or without a filter.

Another common solution involves post-filtering the results using a two-stage approach. The first stage involves retrieval with a large $k$ using an approximate $k$-NN algorithm such as HNSW, while the second stage subselects those results that satisfy the filter criteria using a pre-built index or BM25. These two-stage retrieval systems are limited by the Recall@$k$ of the first stage retriever. We demonstrate that our method significantly improves recall during the first stage retrieval of HNSW for semantic search with filters. Other $k$-NN algorithms, such as Filtered-DiskANN Gollapudi et al. (2023) and FAISS filtering Johnson et al. (2019), either necessitate rebuilding the document index, require constructing new indices that independently encapsulate filter information, or increase query latency by dynamically checking filter criteria while searching through the vectors. That being said, our method can still be combined with these algorithms, potentially serving as a more efficient entry point in the search index.

To our knowledge, the work most closely aligning with our research has been independently explored within the computer vision community under the concept of *compositional learning* Neculai et al. (2022); Nagarajan & Grauman (2018); Vo et al. (2018); Misra et al. (2017). The task in this domain involves retrieving images with specific attributes, such as *dogs* that are also *large* or *brown*. However, as far as we are aware, these methods are not model-agnostic and/or necessitate joint pre-training.

On the other hand, the emergence of semantically meaningful directions in deep neural networks has been extensively studied, including the seminal work on word vectors by Mikolov et al. (2013). Recent advancements in mechanistic interpretability have further uncovered deep connections between feature representations (i.e., filters) and neuron activations (i.e., vector directions) Bricken et al. (2023); Cammarata et al. (2020). Previous studies have employed linear probes to uncover syntax trees and linguistic features within transformer models Hewitt & Manning (2019); Coenen et al. (2019). Our method is inspired by these investigations, aiming to leverage the geometry of these models using linear probes for the downstream task of semantic search with filters.

## 3 METHOD

**Problem** The problem we study is as follows. We are given a corpus of $n$ documents $D$, their corresponding filters, and corresponding embedding vectors $V_D$ as generated by a transformer model. For a given corpus, the collection of filter sets $\{F_1, F_2, \cdots\}$ is fixed and we denote it as $\mathcal{F}$. For each filter set $F_a \in \mathcal{F}$, every document $d_i \in D$ possesses some filter $f_a^i \in F_a$. A document is said to possess a filter if it satisfies that particular filter criteria. The task, for a given query $q$ with filter $f$ in some $F_a$, is to retrieve $k$ documents that are relevant to $q$ and satisfy $f$.

For example, if the corpus $D$ represents a shoe catalog, the collection $\mathcal{F}$ of filter sets could be {brand, color, size, ...}, where the filter set $F_1$ for brand may include {*Nike, Sony, Apple, ...*}. Given a query, $q = $ *athletic shoe* with filter $f = $ *Nike*, we wish to find $k$ items from the catalog semantically matching the query *athletic shoe* and exactly matching the filter *Nike*.

We assume that $|F_a| \ll |D|$ for all $a$, e.g., the number of distinct brands is much smaller than the size of the corpus. Additionally, we assume that each document possesses a unique filter for a given filter set, i.e., a document corresponding to an athletic shoe cannot possess both *Nike* and *Puma* as filters (but can possess *Nike* and *purple*, since they belong to different filter sets).

**Intuition** To the best of our knowledge, efficiently filtering a $k$-NN vector search remains an open combinatorial geometry problem, even for the special case of retrieving documents containing a specific string or keyword. One natural approach toward filtered $k$-NN search would be to limit

the search solely to the subregion of vectors meeting the filter criteria. For an arbitrary distribution of vectors, this region could be highly non-linear, disconnected, and challenging to find. The core premise of our method is the observation that vector embeddings generated by transformers possess considerable linear structure, for reasons not yet fully understood, allowing for the identification of semantically meaningful directions.

To be precise, we propose that every semantically meaningful filter can be associated with a corresponding direction, and that the documents matching these filters are those whose embeddings have a high dot product with this direction. We hypothesize that these directions exist either because a sufficiently powerful language model can implicitly infer a connection between documents and their filters (for example, drawing a connection between *Air Jordans* and the filter *Nike*), or because the document explicitly mentions the filter in its description (*Volkswagen Beetle* and the filter *Volkswagen*), and the model has contextualized it within the document's context.

To build our intuition about these embeddings, let us first understand the behavior of random vectors in high dimensions. Unless otherwise specified, we always work with unit norm vectors. Therefore, the dot product between two vectors and the angle between them can be used interchangeably. The dot product of two random unit vectors in $d$ dimensions has an expectation of $0$ and standard deviation $1/\sqrt{d}$; this underlies the well known fact (e.g. Arora (2014)) that random vectors in high dimensions are approximately perpendicular and make an angle of roughly $\theta \sim (90 \pm \alpha_d)°$, where $\alpha_d \approx \arcsin(1/\sqrt{d})$. For $d = 384$, as in the popular search transformer model `minilm`, this puts the majority of vectors with in the $87°$–$93°$ range of each other, and we approach exact orthogonality in the limit where $d \to \infty$.

However, does this hold true for vectors output by search transformers? To investigate this, we select random documents from the widely-used MS Marco Bajaj et al. (2018) dataset and use `minilm` to vectorize the documents. We compute the pairwise angles between all documents in Figure 2 left. We find that the distribution of angles is concentrated near orthogonality with a mean of approximately $91°$. For unit vectors, this is equivalent to vectors spreading isotropically on the unit sphere. A similar result holds true for two other popular search transformers that we consider, `mpnet` and `sgpt`.

It is worthwhile to contrast this with literature on anisotropic embeddings in pre-trained transformer models like GPT-2 and BERT (Ethayarajh, 2019; Timkey & van Schijndel, 2021). Anisotropy has been linked to poor performance and representation degeneration, prompting proposals for isotropic methods (Jung et al., 2022; Rajaee & Pilehvar, 2021; Gao et al., 2019; Liang et al., 2021). A priori, a pre-trained model could have anisotropic word embeddings yet excel at language modeling, given that next token prediction depends on hidden states across all layers and past tokens. However, for transformers used for search, the geometry of the embeddings directly impact $k$-NN search and anisotropy could degrade performance. Fortunately, search transformers are fine-tuned using contrastive loss, and we empirically find that they are isotropic and thus utilize the entire unit sphere uniformly – the most information-theoretically appealing solution (Wang & Isola, 2020; Elhage et al., 2022). See Appendix E for more discussion.

Given that MS Marco is part of the training distribution of `minilm`, we also conduct the experiment with the out-of-distribution Amazon ESCI shopping dataset Reddy et al. (2022). The dataset consists of e-commerce product descriptions with other attributes such as product brand and color. We treat the product descriptions as documents, and vectorize them while the brands and colors serve as filter sets $F_a$. As shown in Figure 2 left, we find that the distribution, although not centered around $(90 \pm \alpha_d)°$, is still close to being orthogonal.

On the other hand, if we select documents conditioned on a particular filter, e.g., only selecting *Nike* products from the dataset, we find that the vectors are more correlated and make an angle $\theta$ much smaller than $(90 - \alpha_d)°$, as shown in Figure 2 left (although the variance is high). On the unit sphere this corresponds to all *Nike* document vectors making a cone with an opening angle $\theta$. This allows us to learn a vector direction that has a large dot product, i.e., small angle with documents possessing a particular filter while being almost orthogonal to the rest of the documents. Indeed, for the Amazon ESCI dataset, we show in Figure 2 right that we can learn a direction vector for *Nike* such that it exhibits a high dot product with *Nike* items and a low dot product with products associated with different filters, such as *Sony, Puma, Adidas*. While we use the algorithms mentioned below to learn the filter direction, a simple mean pooling of all *Nike* document vectors could also be used as the filter direction for *Nike*, since mean pooling simply reproduces the axis around which the *Nike* document

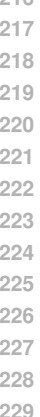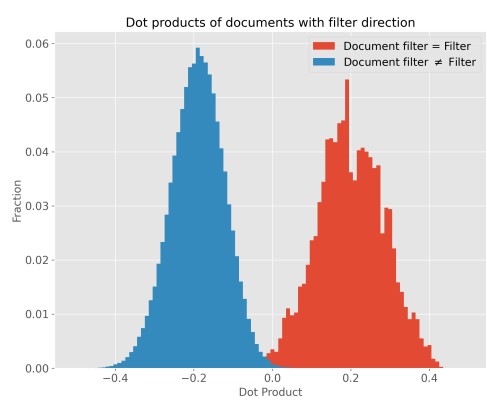

Figure 2: Left: Vector embeddings of randomly selected documents, shown in red and blue, are almost orthogonal to each other. While vector embeddings of documents that have the same filter (*Nike*), shown in purple, are more collinear to each other. Right: Histograms of the dot products between a specific filter direction and document vectors. The dot products between the learned filter direction for *Nike* and *Nike* documents, shown in red, are significantly higher than those between the learned filter direction for *Nike* and random documents, such as *Apple, Puma* and *Adidas*, shown in blue. This separation of scales could thus be utilized to find documents that have a particular filter. All vectors are unit norm.

vectors form the cone. For all experiments, the filter directions are trained and tested on separate splits of the dataset.

However, it is crucial to acknowledge that not every filter can be accurately represented by a vector direction. In Appendix D.3, we evaluate how the fidelity of vector directions to filters diminishes as the semantic meaning of the filter decreases. As an extreme example, if filters are random alphanumeric strings, we fail to identify a vector direction that outperforms random chance at selecting documents that belong to a particular filter. This occurs because documents with the same alphanumeric filter are nearly orthogonal to each other, preventing us from learning a vector direction that has a high dot product exclusively with documents possessing that filter. In other words, document vectors with a particular alphanumeric filter form a cone with opening angle $\approx 90°$, i.e., a degenerate flat cone.

**Algorithm**   We now describe a pair of algorithms. Algorithm 1, run independently for each filter set $F_a \in \mathcal{F}$, attempts to learn a collection of linear probes (i.e. a collection of directions in the space) that best predicts which filter $f$ in $F_a$, if any, applies to a given document. To turn this classification problem into a geometric one, we use a one-hot encoding to identify each filter with a basis direction in $\mathbb{R}^{|F_a|}$, and learn a linear mapping from these one-hot encoded filter vectors into the space of document vectors. This mapping is constructed to maximize the inner product between the learned filter directions and the embedded documents matching the corresponding filter. The mapping of these vectors, which define the columns of the learned matrix $R_{F_a}$, will form the set of linear probes we use throughout the remainder of the paper.

---

**Algorithm 1** Linear probe

---

1: **Input** document vectors $V_D \in \mathbb{R}^{n \times d}$, filters $f_a^i \ \forall i \in \{1, 2, \ldots, n\}$, filter set $F_a \in \mathcal{F}$.

2: Encode each document filter $f_a^i \in F_a$ using a one-hot row vector of dimension $|F_a|$. Stack them to create a filter matrix $\mathbf{F_a} \in \mathbb{R}^{n \times |F_a|}$.

3: Initialize $R_{F_a}$ from $\mathcal{U}\left(\frac{-1}{\sqrt{d}}, \frac{1}{\sqrt{d}}\right)^{d \times |F_a|}$, and use gradient descent to minimize mean squared error loss $\mathcal{L} = ||V_D \cdot R_{F_a} - \mathbf{F_a}||_2^2$

4: Normalize the columns of $R_{F_a}$ to have unit norm, i.e., $\Sigma_{i=1}^d \left(R_{F_a}\right)_{ij}^2 = 1, \forall j \in \{1, 2, \ldots, |F_a|\}$

5: **Output** matrix of directions $R_{F_a}$.

---

While other techniques could be used to find $R_{F_a}$, such as solving $V_D \cdot R_{F_a} - \mathbf{F_a} = 0$ directly using pseudo-inverses, the speed, numerical stability, and extensibility of gradient descent prove very helpful. Since we work with transformers with unit norm embeddings, we explicitly normalize the filter directions to have unit norm in step 4 of the algorithm. We also find that using a cross entropy loss instead of mean squared loss in step 3 yields equally good results.

Algorithm 2 shows how we can apply these linear probes at query time to retrieve a subset of filtered, semantically relevant documents. Given such an $F_a$ and the matrix $R_{F_a}$ learned for $F_a$ in Algorithm 1, we apply the same one-hot-encoding scheme to $f$ in order to identify the index of the column of $R_{F_a}$ which forms its corresponding filter direction $\vec{\nu}_f$. We then add a scaled version of $\vec{\nu}_f$ to the model embedding of the concatenated query and filter strings $\vec{v}_{q+f}$, and output the normalized result.

---

**Algorithm 2** Query-time filter

---

1: **Input** model embedding $\vec{v}_{q+f} \in \mathbb{R}^d$, filter set $F_a$, query filter $f \in F_a$, probe matrix $R_{F_a}$, $\lambda$

2: Let $i_f$ be the index of filter $f$ in the one-hot encoding of $F_a$, as computed in Algorithm 1.

3: $\vec{\nu}_f \leftarrow$ column $i_f$ of $R_{F_a}$

4: $\tilde{v}_q \leftarrow \vec{v}_{q+f} + \lambda\vec{\nu}_f$

5: **Output** $\tilde{v}_q/|\tilde{v}_q|$: combined normalized vector

---

At query time, the vector generated by Algorithm 2 is passed to the $k$-NN search algorithm to conduct semantic search in the presence of a filter. The weight factor $\lambda$ is a hyper-parameter that needs to be tuned using a held-out supervised test set of (queries, filters, documents) triples. Slight variations on the combination strategy in Line 4 also work. In Appendix D.1, we consider the combinations $(1 - \lambda)\vec{v}_q + \lambda\vec{\nu}_f$ and $(1 - \lambda)\vec{v}_{q+f} + \lambda\vec{\nu}_f$ and find that it yields comparable performance.

A priori, the existence of a filter direction does not necessarily imply that the nearest neighbors of a linear combination of the query and filter vector will yield relevant results. In principle, the nearest neighbors could all be completely unrelated to the query and the filter. In the following section, we prove that, with high probability and under mild assumptions, this is not the case.

## 4 THEORY

The quality of the results coming out of our $k$-NN search depends on the number of *irrelevant* versus *relevant* search results we return among the top $k$ nearest vectors. Intuitively, irrelevant results are those items that have little to do with the query ("D-cell battery" as a response to "sweater, red"), and therefore the process generating their corresponding vector is independent of either the choice of the query vector $\vec{v}_q$ or the filter vector $\vec{v}_f$. In contrast, relevant search results are those that directly have to do with the query ("Striped Christmas sweater" as a reply to the same query), and the generative process is consequently a function of some combination of $\vec{v}_q$ and $\vec{\nu}_f$.

Specifically, with respect to fixed i.i.d $\mathcal{N}(\vec{0}, I_d/d)$ samples of a query vector $\vec{v}_q$ and filter vector $\vec{\nu}_f$ (a standard model for random unit vectors of near-unit length in high-dimensional Euclidean space), we assume the data is generated via one of two generative processes:

- *Irrelevant vectors* are sampled from some spherical Gaussian of mean $\vec{0}$ and covariance $\frac{1}{d}I_d$, independent of $\vec{v}_q$ and $\vec{\nu}_f$.

- *Relevant vectors* are sampled from some spherical Gaussian of covariance $\frac{\sigma^2}{d}I_d$ centered around $\vec{v}_q + \gamma\vec{\nu}_f$ for some unknown $\gamma, \sigma \in [0, 4/5]$.

Crucially, we do not assume that we know the "true" value of the parameter $\gamma$. Instead, running Algorithm 2 with some $\lambda$ of our choosing will produce (a normalized version of) $\vec{v}_q + \lambda\vec{\nu}_f$ around which we search for the $k$ nearest neighbors[1]. We argue that this seemingly "wrong" query, with $\gamma$ replaced by $\lambda$, will still be overwhelmingly likely to find relevant vectors nearby rather than irrelevant ones. Note that, as mentioned in Section 3, our analysis in this section depends on the query vector $\vec{v}_*$ being a function of $\vec{v}_q$ rather than the less interpretable $\vec{v}_{q+f}$ which gives experimentally similar results but cannot be nicely decomposed into simpler components.

---

[1]For simplicity, we omit these normalizing factors from our analysis, which readily cancel out in all key calculations.

These two distributions model the distribution of vectors previously discussed in Section 3, and elaborated on in Appendix E. The i.i.d. Gaussian distribution from which we sample "irrelevant" vectors matches a well-studied model of isotropic vectors on a high-dimensional sphere (e.g. Vershynin (2018)), and the distribution of relevant vectors is an attempt to model the conical distribution we discover for vectors that match semantically meaningful filters.

While it is challenging to validate these assumptions by directly establishing the Gaussianity of an arbitrary distribution, in Figure 3, we plot the distribution of irrelevant vectors and relevant vectors around $v_q$, for a random vector component. We find that it closely follows a Gaussian distribution with standard deviations in the required range. In Appendix E, we provide Q-Q plots that further corroborate this claim, and contrast our findings with those on anisotropy in the literature.

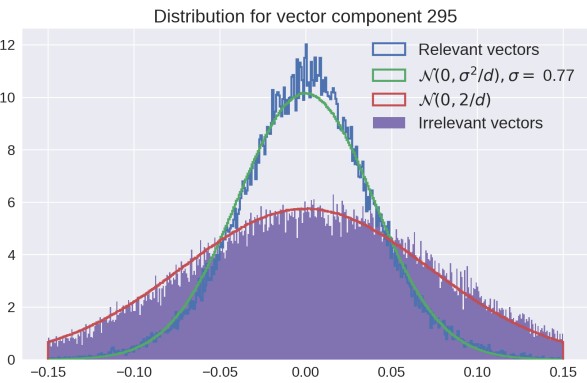

Figure 3: Distribution of vector components generated by `minilm` on the Brands dataset

As a warmup, we build intuition for the efficacy of Algorithm 2 in this setting by first considering the (somewhat degenerate) case where $\gamma = \lambda = 0$. This corresponds to ordinary vector search without any filters. In this special case, the assumptions imposed above stipulate that for any sample of the combined query-filter vector $\vec{v}_q + \gamma\vec{\nu}_f = \vec{v}_q$ and irrelevant vector $\vec{v}_i$, the difference $\vec{v}_q - \vec{v}_i$ is distributed as $\mathcal{N}\left(0, \frac{2}{d}I_d\right)$. Conversely, the difference with a relevant vector $\vec{v}_r$ is distributed as $\mathcal{N}\left(0, \frac{\sigma^2}{d}I_d\right)$. Because $\frac{\sigma^2}{d} \leq \frac{(4/5)^2}{d} < \frac{2}{d}$, squared distances to relevant vectors will typically be much smaller than those to irrelevant vectors, ensuring relevant $k$-NN results with high probability. This separation of scale in these variances could be attributed to neural network training, where the objective *is* to bring relevant vectors closer together and push irrelevant vectors further apart in the embedding space. In the more general case where $\gamma$ and $\lambda$ can vary, the gap will depend non-linearly on the specific choice of those values. Details are left to the proof of Theorem 1.

Under these assumptions, it is fairly intuitive that relevant vectors have a definite advantage in terms of likelihood of being nearest neighbors of our modified query $\tilde{v}_q$, especially when $\sigma$ is small and $\lambda$ is close to $\gamma$. However, when working with large datasets, it is plausible that the sheer size of the dataset $D$ may overwhelm this advantage by containing so many irrelevant samples that many of them end up as $k$ nearest neighbors by accident.

Our analysis shows that for reasonable settings of parameters, this is not the case. We take advantage of the high dimensionality of the vector spaces we work with, and show that the probability that any given irrelevant vector is among the $k$-nearest neighbors decreases exponentially in $d$. Intuitively, this happens because each additional dimension gives more room for an irrelevant vector to be far from $\tilde{v}_q$ than it does for relevant vectors. By taking the union bound over all vectors of the dataset, we show that the expected number of irrelevant results among the $k$-nearest neighbors is oftentimes much less than 1. This statement is captured in the following theorem.

**Theorem 1.** *Let $\vec{v}_q$ and $\vec{\nu}_f$ be independent $\mathcal{N}(\vec{0}, I_d)$ samples representing the embedding of some query $q$ and filter $f$. Let $\lambda, \gamma, \sigma \in [0, 4/5]$. Let $D \subset \mathbb{R}^d$ be a dataset of $n$ embedded passage vectors irrelevant to $\vec{v}_q$ and $\vec{\nu}_f$, and let $R$ be a set of relevant vectors sampled as defined above. As long as $|R| \geq k$, the expected number of irrelevant vectors among the $k$ nearest neighbors of $\vec{v}_q$ is at most $0.943^{d-1}kn$.*

In the analysis, we analyze two different distributions: that of squared distances between the query vector $\vec{v}_* = \vec{v}_q + \lambda \vec{\nu}_f$ and irrelevant vectors ($\mathcal{D}_r$), and that of the squared distances between $\vec{v}_*$ and relevant vectors ($\mathcal{D}_i$). We show that these are both scaled mostly-independent $\chi^2(d)$ distributions, and that the probability that $\vec{v}_*$ is closer to a vector drawn from $\mathcal{D}_r$ than one from $\mathcal{D}_i$ matches the probability that the difference of two other closely related (but now truly independent) $\chi^2$ distributions is negative. By analyzing the moment generating function of this difference (Lemma 2), we show that it only takes on negative values an exponentially small fraction of the time.

**Lemma 2.** *Let $X_1$ and $X_2$ be independent random variables sampled as $X_1 \sim c_1 \chi^2(k)$ and $X_2 \sim c_2 \chi^2(k)$, respectively. If $c_1 > 2c_2$, then $\Pr[X_2 \geq X_1] \leq \sqrt{8/9}^k \approx 0.943^k$.*

Given Lemma 2 (whose proof is deferred to Appendix A), one can derive Theorem 1 as follows.

*Proof of Theorem 1.* Let $\vec{v}_r$ be the embedding of a vector relevant to both $q$ and $f$, and let $\vec{v}_i$ be the embedding of a vector irrelevant to both. Define $\mathcal{D}_i$ as the distribution over $||\vec{v}_* - \mathcal{N}(\vec{0}, I_d)||_2^2$, i.e. the prior distribution of the squared distance from some fixed $\vec{v}_*$ to an irrelevant $\vec{v}_i$. Define $\mathcal{D}_r$ as the distribution over $||\vec{v}_* - (\vec{v}_q + \gamma \vec{\nu}_f + \mathcal{N}(\vec{0}, \sigma^2 I_d))||_2^2$, i.e. the prior distribution of the squared distance from a fixed $\vec{v}_*$ to a relevant $\vec{v}_r$.

We can evaluate $\mathcal{D}_r$ and $\mathcal{D}_i$ as follows

$$\mathcal{D}_r = ||\vec{v}_* - (\vec{v}_q + \gamma \vec{\nu}_f + \mathcal{N}(\vec{0}, \sigma^2 I_d))||_2^2 = ||\vec{v}_q + \lambda \vec{\nu}_f - (\vec{v}_q + \gamma \vec{\nu}_f + \mathcal{N}(\vec{0}, \sigma^2 I_d))||_2^2$$

$$= ||\mathcal{N}(\vec{0}, \sigma^2 I_d) + (\lambda - \gamma)\mathcal{N}(\vec{0}, I_d)||_2^2 = (\sigma^2 + (\lambda - \gamma)^2)\chi^2(d),$$
$$\text{and}$$
$$\mathcal{D}_i = ||\vec{v}_* - \mathcal{N}(\vec{0}, I_d)||_2^2 = ||(\mathcal{N}(\vec{0}, I_d) + \lambda \mathcal{N}(\vec{0}, I_d)) - \mathcal{N}(\vec{0}, I_d)||_2^2$$

$$= ||\mathcal{N}(\vec{0}, (2 + \lambda^2)I_d)||_2^2 = (2 + \lambda^2)\chi^2(d)$$

Thus, both distributions are scaled $\chi^2$ distributions with the same parameter $d$. However, having fixed a value for $\vec{v}_*$, the squared distances in $\mathcal{D}_r$ and $\mathcal{D}_i$ both depend on our choice of $\vec{v}_*$, so jointly sampling $x_1 \sim \mathcal{D}_r$ and $x_2 \sim \mathcal{D}_i$ for the same $\vec{v}_*$ yields random values that are not entirely independent. However, rotating the sampled vectors so that $\vec{v}_*$ lies on a basis direction shows that the samples are still independent among the remaining $d - 1$ dimensions, and hence $x_1 - x_2$ can be expressed as the difference of two $\chi^2(d - 1)$ distributions with scales as above.

$$\mathcal{D} = (2 + \lambda^2)\chi^2(d - 1) - (\sigma^2 + (\lambda - \gamma)^2)\chi^2(d - 1)$$

It can be verified that the ratio $(\sigma^2 + (\lambda - \gamma)^2)/(2 + \lambda^2)$ is at most $1/2$. Therefore, plugging this difference of distributions into Lemma 2, we conclude that the probability it is negative (and hence the probability that a sample from $\mathcal{D}_r$ exceeds that from $\mathcal{D}_i$) is at most $0.943^{d-1}$.

The above statement is true for a single sample from $\mathcal{D}_i$ and $\mathcal{D}_r$. With $k$ samples from $\mathcal{D}_r$, the probability of a single sample from $\mathcal{D}_i$ exceeding any of the $k$ of them (and thus appearing among the $k$ nearest neighbors) is at most $0.943^{d-1}k$. By a second union bound, the probability that of the $n$ samples in $D$ is one of the $k$-nearest neighbors is at most $0.943^{d-1}nk$. ∎

For a 384-dimensional dataset (such as `minilm`) supported on 1B points, this calculation estimates that at most $1.73$ irrelevant vectors will show up in the top $10$ nearest neighbors, or that $80+\%$ of results will be relevant. For higher dimensional models such as the 768-dimensional `mpnet`, the dataset will need to be more than a billion times larger before we expect a single irrelevant result.

## 5 EXPERIMENTS

**Datasets**    To evaluate our method we require a dataset that has queries, passages and filters, although we do not need queries for training. Popular search relevancy datasets such as MS Marco have queries and passages but no filters. To this end, we utilize the Amazon shopping queries ESCI dataset Reddy et al. (2022) mentioned in earlier sections. The ESCI dataset contains real e-commerce user queries, Amazon products, and ground truth relevancy ratings. Each product comes with several attributes such as product description, brand, color, etc. We sub-sample the dataset to only contain products that exactly match some query.

We create three different evaluation datasets using the `English (US) test` split: two real datasets based on *colors* and *brands*, and one synthetic dataset based on manufacturing *countries*. For the former, we first find the top 40 and 100 frequent colors and brands respectively, and only keep products that satisfy this criteria. For the latter, we create a synthetic attribute for each product item by sampling a {country} from a random list of 50 countries and correspondingly concatenate "Made in {country}" to the product description. This concatenation is crucial for the *countries* dataset since the product description otherwise has no information about the {country}, and it would be impossible to learn a filter direction. For the real datasets, no such concatenation is performed. All of our datasets are always balanced, ensuring that the filters are uniformly distributed across both the training and evaluation dataset. These constraints significantly change the retrieval task from Reddy et al. (2022), and thus our numbers cannot be compared directly to the ESCI leader board.

For each relevant query-product pair $(q_i, d_i)$, we form a triple $(q_i, d_i, f_i)$, where $f_i$ represents the associated product color, brand, or manufacturing country of that product, $d_i$ is the product description and $q_i$ is the user query. During evaluation, we simulate a scenario where each query $q_i$ is accompanied by a filter $f_i$, and $d_i$ is the intended result. Dataset statistics can be found in Table 1 where, `qrels` refers to the total number of $(q, d, f)$ triples and $|F|$ refers to the total number of different filters for that evaluation dataset. In Appendix C, we provide dataset examples, list of filters, and statistics about textual overlap between product descriptions and filters.

Table 1: Details of the evaluation datasets

| Dataset | docs | queries | qrels | $|F|$ |
|---|---|---|---|---|
| Colors | 40,072 | 1,581 | 2,379 | 40 |
| Brands | 61,600 | 1,411 | 3,974 | 100 |
| Countries | 12,500 | 11,247 | 12,500 | 50 |

**Models and training**   For our experiments, we employ three different pre-trained search transformers: `minilm`, `mpnet`, and `sgpt`, using the Sentence Transformers library Reimers & Gurevych (2019); Muennighoff (2022). We work with pre-trained models to emulate the real world scenario where search queries are often unavailable for fine-tuning. Indeed, our algorithm does not require queries for learning the filter direction. The models have 22M, 110M, and 1.3B parameters, with dimensionalities of 384, 768, and 2048 respectively. Their performance on the original Task 1 ESCI challenge, as measured in terms of nDCG@10, is 0.54, 0.53, and 0.53, respectively. The first two models produce unit norm vectors, and the latter, although not inherently outputting normalized vectors, was fine-tuned with cosine similarity. This allows us to explicitly normalize its embeddings without affecting the performance[2].

All models are utilized with frozen weights, and the linear probe is trained using the `English (US) train` split of the Amazon ESCI dataset for each model and dataset. Like for the evaluation dataset, we curate three different train datasets and keep them balanced. The training dataset consisted of 150, 70, and 2000 instances for each filter for *colors*, *brands*, and *countries* respectively, resulting in total training dataset sizes of 6K, 7K, and 100K, respectively. We utilized Adam with a learning rate ranging from $1 \times 10^{-1}$ to $1 \times 10^{-2}$, employing lower learning rates for the `sgpt` model and the *countries* dataset. The learned linear probes, on a held-out set, on average show an accuracy of 0.50, 0.99 and 0.82 the three datasets respectively. More details on training and performance of linear probe, and the optimal values of $\lambda$ can be found in Appendix B.

**Results and baselines**   We compare our method against two related, motivated baselines satisfying criteria key of our work: they must neither substantially affect query latency, nor may they alter the $k$-NN index. The first baseline involves concatenating the query and the filter, denoted by $q + f$, and sending the model embedding of the concatenated string $\vec{v}_{q+f}$ to $k$-NN search, effectively treating it as our method, $\vec{v}_{q+f} + \lambda \vec{v}_f$, with $\lambda = 0$. Performance is measured in our terms of nDCG@10, a standard metric for evaluating the accuracy of search results. The results can be found in Table 2 left. On average we find a boost of **15.3%**, **22.0%** and **25.8%** for `minilm`, `mpnet` and `sgpt` respectively. The performance on *colors* is the least impressive of all, presumably because many product descriptions do not explicitly mention the color, and color names such as *black*, *white*, or *blue* are more ambiguous than brand names like *Nike* or *Sony*. This ambiguity makes it more challenging to identify a unique

---

[2]While theoretical analysis formally assumes vectors are drawn from a spherical Gaussian, the concentration of norms in high dimensional Gaussians makes the distinction between the cases minimal.

Table 2: Our method vs baseline for different models and datasets in terms of nDCG@10 and Recall@10.

| | nDCG@10 | | | Recall@10 | | |
| | Colors | Brands | Countries | Colors | Brands | Countries |
|---|---|---|---|---|---|---|
| minilm baseline | 0.2328 | 0.4194 | 0.4522 | 0.3710 | 0.5477 | 0.6362 |
| minilm ours | **0.2609** | **0.4713** | **0.5496** | **0.4031** | **0.6177** | **0.7172** |
| mpnet baseline | 0.2316 | 0.4088 | 0.5033 | 0.3667 | 0.5334 | 0.7054 |
| mpnet ours | **0.2797** | **0.4962** | **0.6230** | **0.4335** | **0.6392** | **0.7982** |
| sgpt baseline | 0.2294 | 0.4293 | 0.4644 | 0.3299 | 0.5582 | 0.6771 |
| sgpt ours | **0.2749** | **0.4831** | **0.6733** | **0.4080** | **0.6354** | **0.8351** |

direction corresponding to the filter within the context of the search corpus, as is also reflected in the performance of the linear probe in Table 3 in Appendix B.

In the above experiment, we employ exact $k$-NN search to isolate and quantify the accuracy benefits of our method. This approach avoids the potential variability introduced by approximate $k$-NN search methods. Conversely, our second baseline is designed to measure the effectiveness of our method in real-world systems, which typically employ approximate $k$-NN algorithms such as HNSW. We refer to this baseline as post-filtering. This involves two steps: first, executing the query using approximate $k$-NN search to retrieve an initial set of $k'$ results, and second, applying a post-filtering step to retain only the top $k$ documents that satisfy the filter criteria. Since we're working with an expanded initial result set, the appropriate comparison metric is Recall@$k'$. The results for $k' = 10$ can be found on Table 2 right. We choose a relatively small value of $k'$ because, on average, each query in the dataset corresponds to approximately one relevant document and filter, and we are operating within a relatively small corpus. We find an average boost of **11.4%, 17.1%** and **20.3%** for the three models. Results for $k' = 100$ and HNSW implementation details can be found in Appendix D.

Figure 4: Performance for the task of double filters, and its sensitivity to $\lambda$ for single filters

| | Colors & Countries | Brands & Countries |
|---|---|---|
| minilm baseline | 0.5068 | 0.6167 |
| minilm ours | **0.6283** | **0.8045** |
| mpnet baseline | 0.5189 | 0.6403 |
| mpnet ours | **0.6946** | **0.8421** |
| sgpt baseline | 0.4892 | 0.6212 |
| sgpt ours | **0.7480** | **0.8890** |

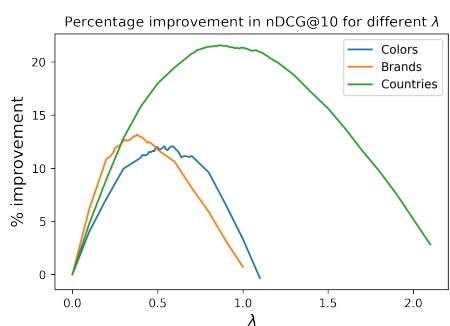

**Other experiments and ablations** We also assess the performance of our method when employing two different filter sets together, such as colors and brands. We now send $\vec{v}_{q+f_c+f_b} + \lambda_1 \vec{\nu}_{f_c} + \lambda_2 \vec{\nu}_{f_b}$ to $k$-NN search, where each $\lambda_i$ is a separate hyperparameter and $\vec{\nu}_{f_c}, \vec{\nu}_{f_b}$ are the probe vectors corresponding to colors and brands respectively. For the corresponding baseline, $\vec{v}_{q+f_c+f_b}$ is sent to $k$-NN search; our method performs significantly better as shown in Figure 4 left. We also study the impact of varying $\lambda$ on search performance as measured. Figure 4 right, shows the effect of varying $\lambda$ for each of the three models for the *brands* dataset. In Appendix D, we conduct additional experiments and ablations. We explore different ways of combining $\vec{v}_q$ and $\vec{v}_f$ in D.1, and find that different linear combinations yield comparable performance. We determine the importance of identifying filter directions using linear probes as opposed to just using the model embedding of the filter in D.2. We also evaluate our method with a pre-trained model that outputs non-normalized embeddings in D.4, and with a fine-tuned mpnet model in D.5.

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

## A   PROOF OF LEMMA 2

Below we include the proof of Lemma 2, left out of the main paper due to space considerations.

**Lemma 2.** *Let $X_1$ and $X_2$ be independent random variables sampled as $X_1 \sim c_1 \chi^2(k)$ and $X_2 \sim c_2 \chi^2(k)$, respectively. If $c_1 > 2c_2$, then $\Pr[X_2 \geq X_1] \leq \sqrt{8/9}^k \approx 0.943^k$.*

*Proof of Lemma 2.* The moment generating functions (MGFs) $M_1, M_2$ of the distributions from which $X_1$ and $X_2$ are respectively sampled can be shown to satisfy

$$M_1(t) = (1 - 2c_1 t)^{-k/2}$$
$$M_2(t) = (1 - 2c_2 t)^{-k/2}$$

and are both valid where $t < .5/\max(c_1, c_2) = .5/c_1$. Therefore, the MGF $M$ of $X = X_1 - X_2$ takes the form

$$M(t) = M_1(t)M_2(-t)$$
$$= ((1 - 2c_1 t)(1 + 2c_2 t))^{-k/2}$$

valid where $|t| < 1/(2c_1)$ and in particular for $t = -1/(4c_1)$. By a generic Chernoff bound for moment generating functions (see e.g. Mitzenmacher & Upfal (2017)),

$$\Pr[X_2 \geq X_1] = \Pr[X_1 - X_2 \leq 0]$$
$$\leq \inf_{t \in (-.5/c_1, 0)} M(t)$$
$$\leq M(-1/(4c_1))$$
$$= ((3/2) \cdot (1 - c_1/(2c_2)))^{-k/2}$$
$$\leq ((3/2) \cdot (3/4))^{-k/2}$$
$$\leq (8/9)^{k/2}$$

$\square$

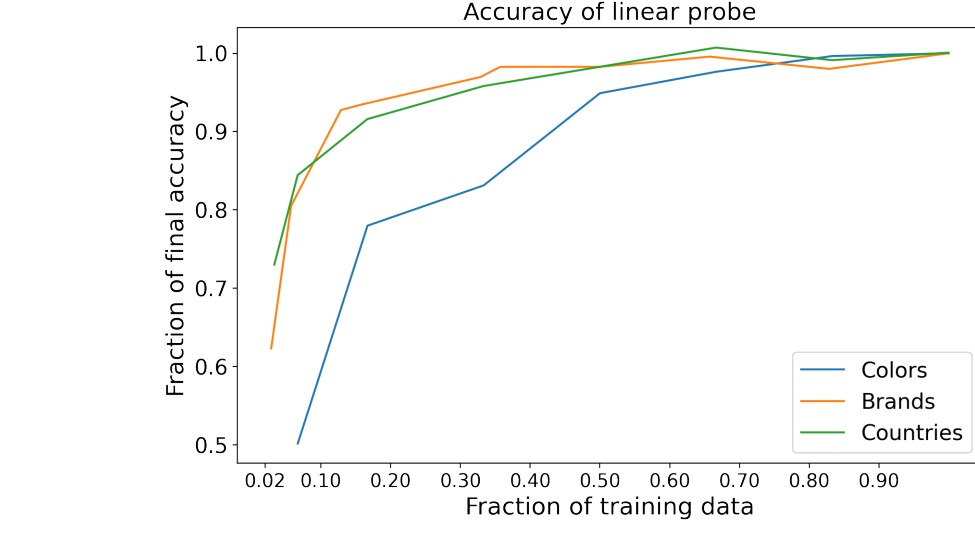

Figure 5: Accuracy of the linear probe for the three datasets for the model `minilm`. The full training data consists of 150, 70, and 2000 samples for each filter type for *colors*, *brands*, and *countries* respectively. The accuracy when trained on the full data is 0.42, 0.99 and 0.74 respectively. The $y$-axis shows the accuracy as a fraction of peak accuracy and $x$-axis shows the fraction of training data used. Results averaged over 5 training runs.

## B    TRAINING DETAILS

In this appendix, we provide additional training details and experiments. For approximate $k$-NN search, HNSW, we use the FAISS implementation with standard parameters, `efsearch` and `efconstruct` parameters of 100 and 128 respectively. The former balances search accuracy against speed, while the latter affects index quality and build time, both through the regulation of candidate list sizes; that is, larger values yield greater accuracy but at the cost of increased latency.

As mentioned above, the training dataset consisted of 150, 70, and 2000 instances for each filter type for *colors*, *brands*, and *countries* respectively, resulting in total training dataset sizes of 6K, 7K, and 100K, respectively. We found that gradient descent converges rapidly, typically within a few thousand iterations, requiring less than a quarter hour of wall clock time on a single Tesla V100 GPU for each run. The hyperparameter $\lambda$ for each run is optimized by evaluation on a held-out set from the `train` split. We document the accuracy of the linear probe on the held-out set for each dataset and model in Table 3.

| Dataset | minilm | mpnet | sgpt |
|---|---|---|---|
| Colors | 0.42 | 0.52 | 0.51 |
| Brands | 0.99 | 0.99 | 0.99 |
| Countries | 0.74 | 0.79 | 0.92 |

Table 3: Accuracy of the linear probe on a held-out set.

| Dataset | minilm | mpnet | sgpt |
|---|---|---|---|
| Colors | 0.59 | 0.91 | 0.99 |
| Brands | 0.38 | 0.41 | 0.46 |
| Countries | 0.86 | 0.78 | 1.5 |

Table 4: Optimal value of $\lambda$ for each model and dataset.

The optimal value of $\lambda$ for each model and dataset is obtained by evaluating on a held out set and can be found in Table 4. We also measure the performance for the linear probe as we change the number of training samples per filter in Figure 5 for the model `minilm` for all three datasets.

## C    DATASET STATISTICS

In this appendix, we present explicit examples from the Amazon ESCI dataset for the reader's convenience and explore the textual overlap between the filters and the documents. For training the linear probe and evaluating the nDCG and Recall metrics, we consistently utilize a balanced dataset, ensuring that the filters are uniformly distributed across both the training and evaluation datasets.

Table 5 shows some representative samples from the evaluation datasets, and the set of filters for each filter category is given below.

| Filter | Query | Document |
|---|---|---|
| Blue | 18x34 pool cover | Swimline S1834OV 18 Foot x 34 Foot Deluxe Above Ground Swimming Pool... |
| Brushed Nickel | l l brackets without screws | YUMORE L Bracket, 5" x 3" Max Load: 35lb/15KG Heavy Duty Stainless Steel Solid Shelf Support Corner Brace Joint Right Angle... |
| Sony | 32 inch smart tv 4k | Sony 32-inch 720p Smart LED TV (KDL32W600D, 2016 Model) X-Reality PRO for a Cleaner, More Refined Picture ... |
| Levi's | brown belt without buckle | Levi's Men's Cut to Fit 3 Pack Web Belt with Buckle, Black/Olive/Khaki, One Size STURDY STRAP: This belt strap is made ... |
| Greece | 100w solar panel without controller | Made in Greece. Renogy 100W Eclipse Lightweight Suitcase Without Controller, Panel Only, Black High-efficiency Solar cells ... |
| Turkey | 1.50 mens xl reading glasses without nose piece | Made in Turkey OLOMEE Reading Glasses 3.0 Oversize Large Square Men Readers 4 Pack,Comfort Lightweight ... |

Table 5: Examples of filters and documents from the Amazon ESCI dataset for *colors*, *brands* and *countries*.

**Colors:** black, white, multicolored, blue, gray, red, silver, green, clear, pink, brown, yellow, gold, purple, orange, beige, stainless steel, navy, rose gold, natural, transparent, dark gray, charcoal, matte black, chrome, tan, navy blue, ivory, bronze, light blue, khaki, burgundy, oil-rubbed bronze, royal blue, light gray, espresso, warm white, dark brown, teal, brushed nickel.

**Brands:** Nike, adidas, Under Armour, Amazon, Amazon Basics, SAMSUNG, Apple, Hanes, Sony, LEGO, Amazon Essentials, Disney, Carhartt, Simple Joys by Carter's, Zinus, Rubie's, HP, Champion, LG, Nintendo, OtterBox, BLACK+DECKER, DEWALT, Crocs, Microsoft, Bose, Hunter Fan Company, Columbia, SweatyRocks, SheIn, Cuisinart, Fender, ASUS, Birkenstock, Panasonic, Gucci, NERF, Michael Kors, Dickies, Levi's, Crayola, DREAM PAIRS, Samsonite, New Balance, Spigen, Romwe, Best Choice Products, Forum Novelties, Hallmark, Fruit of the Loom, Fisher-Price, Safavieh, YETI, Leg Avenue, Oakley, Intex, Nautica, Graco, L'Oreal Paris, Logitech, The North Face, Canon, Anker, Lacoste, Swiffer, KitchenAid, Command, Calvin Klein, SOJOS, STAR WARS, Elite Fan Shop, Avidlove, UGG, Makita, Delta Children, Gildan, mDesign, PAVOI, Tommy Hilfiger, Dell, Ekouaer, VTech, Big Joe, Lee, SPANX, Keurig, Hydro Flask, modelones, Wrangler Authentics, ASICS, Vans, TAG Heuer, Pyle, Acer, Honeywell, amscan, CUPSHE, Allegra K, Polo Ralph Lauren, Fujifilm.

**Countries:** Nigeria, Sweden, New Zealand, Malaysia, Saudi Arabia, Bolivia, Pakistan, Trinidad and Tobago, Taiwan, Iran, Brazil, Philippines, Ghana, Bangladesh, Chile, Vietnam, Japan, Belgium, Thailand, United Kingdom, Greece, Ireland, Italy, Afghanistan, South Africa, Bhutan, Switzerland, Mexico, Netherlands, Egypt, Norway, Turkey, Australia, Poland, Argentina, Qatar, Singapore, Russia, Indonesia, China, South Korea, Spain, Canada, France, India, United Arab Emirates, Germany, Austria, United States, Kuwait.

## C.1 TEXTUAL OVERLAP

We also conduct a study on the overlap between the documents and the filters. Although some product descriptions could implicitly imply the brand—for example, *Air Jordan* would suggest *Nike* and *iPhone* would suggest *Apple*—most products do not exhibit such a strong brand affinity. Given the linear probe's ability to identify relevant filter directions based on document embeddings, it must be the case that most product descriptions explicitly mention the filter name. We find that for *colors* and

*brands*, the product description contains the filter name 77% and 97% of the time, respectively. For the synthetic *countries* data, the overlap is 100%, by construction.

It is worthwhile to mention here that, to the best of our knowledge, even constraining vector search with $k$-NN to only retrieve documents that explicitly contain a string or a keyword remains an open problem.

# D ADDITIONAL EXPERIMENTS AND ABLATIONS

In this appendix we provide additional experiments and ablations to show the efficacy and robustness of our method.

## D.1 IMPACT OF DIFFERENT LINEAR COMBINATIONS

There are several different linear combinations that could be potentially used for implementing the filter criteria. We report some results in Table 6 for the model `minilm`. As mentioned in the main draft, we find that different linear combinations yield comparable performance.

| | Colors | Brands | Countries |
|---|---|---|---|
| $(1-\lambda)\vec{v}_q + \lambda\vec{\nu}_f$ | 0.2592 | 0.4714 | 0.5512 |
| $(1-\lambda)\vec{v}_{q+f} + \lambda\vec{\nu}_f$ | 0.2551 | 0.4663 | 0.6582 |
| $\frac{\vec{v}_{q+f}+\lambda\vec{\nu}_f}{|\vec{v}_{q+f}+\lambda\vec{\nu}_f|}$ | 0.2609 | 0.4713 | 0.5496 |

Table 6: Performance of different linear combination protocols for the model `minilm` in terms of nDCG@10. The last row corrsponds to the protocol used in the main draft.

## D.2 ABLATION STUDY WITH MODEL EMBEDDING

We also conduct experiments where we use the model embedding of the filter instead of finding them using linear probes. That is, we use the transformer output embedding of the word "*Nike*" instead of learning a filter direction. Table 7 shows the results for `minilm`. We find that using linear probes is on average better. In fact, for *colors* and *countries*, the method performs only as good as the baseline.

| | Colors | Brands | Countries |
|---|---|---|---|
| Model embedding | 0.2329 | 0.4443 | 0.4495 |
| Linear probes | 0.2609 | 0.4713 | 0.5496 |

Table 7: Comparison of our method when using linear probes for finding filter directions vs using vector embedding of the filer for the model `minilm` in terms of nDCG@10

## D.3 FILTER FIDELITY WITH ALPHANUMERIC FILTERS

Our method leverages emergent semantically meaningful directions within transformer embeddings. Consequently, its performance diminishes when filters are random alphanumeric strings. To evaluate this, we generated a synthetic dataset of *prices* by appending "\${price}" to documents, where {price} comes from a discrete set of 50 different possible integer values. Contrary to previous instances, where the linear probe could readily identify filter directions associated with various filters, we observed that for *prices*, the probe's accuracy did not exceed random chance. Figure 6 illustrates the dot product of documents with different filter directions, both when they are same and when they differ, highlighting performance comparable to random chance. For comparison, see Figure 2, which depicts the scenario when filters have semantic significance.

## D.4 RESULTS FOR NON-NORMALIZED MODEL `tasb`

In the main draft, we analyze the performance of models that output embeddings of unit norm. This approach is motivated by two main reasons: 1) the most powerful and popular search transformers within the community employ unit norm embeddings, and 2) it ensures better alignment with our theoretical analysis. In this subsection, we measure the performance of a popular transformer model, `tasb`, that outputs non-normalized embeddings Hofstätter et al. (2021). In this scenario, we do not

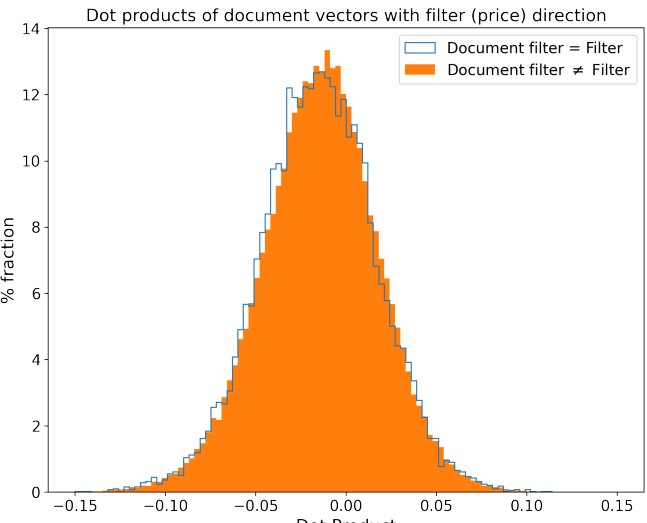

Figure 6: For the numeric filter dataset *prices*, the direction corresponding to a specific filter exhibits similar dot products with all documents, irrespective of the document's filter.

impose unit norm constraints on the filter directions. As illustrated in Table 8, the performance improvement is limited compared to unit-norm models. We do not have a clear understanding of why this is the case and leave this for further investigation.

|  | Colors | Brands | Countries |
|---|---|---|---|
| nDCG@10 baseline | 0.2997 | 0.4575 | 0.5924 |
| nDCG@10 ours | **0.3297** | **0.4890** | **0.6686** |
| Recall@10 baseline | 0.4455 | 0.5879 | 0.7821 |
| Recall@10 ours | **0.4849** | **0.6282** | **0.8286** |

Table 8: Our method vs baseline for a non-normalized model `tasb`.

### D.5 RESULTS FOR THE FINE-TUNED `mpnet` MODEL

We work with pre-trained models in the draft since we want to emulate the real world scenario, where ground truth queries are often unavailable. In this appendix, we show that our method yields impressive performance with fine-tuned models as well. We fined-tuned `mpnet` on the Task 1 train set of Amazon ESCI dataset and ran our experiments for all three filter sets and found the following results for nDCG@10. In Table 9, we present them along with the pre-trained results from the main draft for convenience.

|  | Colors | Brands | Countries |
|---|---|---|---|
| `mpnet` pretrained baseline | 0.232 | 0.409 | 0.503 |
| `mpnet` pretrained ours | **0.280** | **0.496** | **0.623** |
| `mpnet` finetuned baseline | 0.320 | 0.429 | 0.449 |
| `mpnet` finetuned ours | **0.362** | **0.523** | **0.615** |

Table 9: Our method vs baseline in terms of nDCG@10 using the fine-tuned `mpnet` model.

On average, our method with the fine-tuned MPNet model performs 23.5% better than the baseline with the fine-tuned `mpnet` model (line 3 vs. 4). This is comparable to the 22% improvement reported in the main draft for the pre-trained model (line 1 vs. 2).

### D.6 RESULTS ON RECALL@100

We additionally assess the Recall@100 performance for our experiments in Table 10. It is important to note that our evaluation datasets are relatively small, encompassing at most 62K documents, and each query, on average, corresponds to one relevant ground truth result. Consequently, Recall@$k$ for $k = 10$, as reported in the main draft, serves as a more suitable benchmark given the context of our study.

|  | Colors | Brands | Countries |
|---|---|---|---|
| `minilm` baseline | 0.7203 | 0.8880 | 0.8710 |
| `minilm` ours | **0.7351** | **0.9772** | **0.9240** |
| `mpnet` baseline | 0.7546 | 0.8695 | 0.9176 |
| `mpnet` ours | **0.7860** | **0.9809** | **0.9550** |
| `sgpt` baseline | 0.7108 | 0.8930 | 0.9075 |
| `sgpt` ours | **0.7550** | **0.9781** | **0.9621** |

Table 10: Our method vs baseline in terms of Recall@100 using approximate $k$-NN (HNSW).

## E EMPIRICAL EVIDENCE FOR ISOTROPY

In this section, we provide more evidence about the assumptions used in our theoretical analysis. We also study our results on Gaussianity of vectors in context with the results on anisotropy of transformer embeddings found in the literature.

### E.1 GAUSSIANITY OF HIGH DIMENSIONAL VECTORS

Like in Figure 3, in the main draft we plot the distribution of irrelevant and relevant vectors, and $Q-Q$ plots for two different models, `mpnet` and `minilm`, for some randomly selected vector component $j$ in Figure 8 and Figure 9. The models have dimensionalities of 768 and 384 respectively.

### E.2 COMPARISON WITH PRIOR WORK ON ANISOTROPY

It is widely recognized that contextualized word vectors exhibit high cosine similarity across different contexts. For example, the cosine similarity for the word *ground*, whether in the context of *ground truth* or *Earth*, is notably high Ethayarajh (2019). Furthermore, many transformer models feature *rogue* dimensions—specific dimensions that dominate over others in the computation of cosine similarities Timkey & van Schijndel (2021). Such phenomena, characterized by high cosine similarity and the presence of rogue dimensions, have been interpreted as indications of degenerate representations and a lack of expressivity. As a result, numerous studies have attempted to mitigate this anisotropy to enhance model performance Jung et al. (2022); Rajaee & Pilehvar (2021); Gao et al. (2019); Liang et al. (2021). However, the outcomes have been relatively inconsistent, with no isotropization strategy proving to consistently benefit performance Ding et al. (2022).

We delve into these claims from the perspective of our downstream task: search. We posit that the utility of a model's isotropy should be assessed based on its impact on the downstream task. For search applications, anisotropy likely undermines model performance, as the geometry of the vector embeddings directly influences the output. Given that anisotropic embeddings limit the model's capacity to explore the full available space, it is plausible that this restriction could detrimentally affect performance. In essence, geometry has a tangible effect on search performance.

This argument, however, may not extend to models designed for text generation or masked language modeling, where the relationship between geometry and performance is more indirect. Without a thorough mechanistic understanding of how large neural networks function, making definitive claims about these models remains challenging. Either-way we focus on search transformers and find two major themes. First, vector embeddings for search transformers are quite isotropic, and that there do not exist rogue dimensions in the geometry of search transformers that disproportionately affect performance.

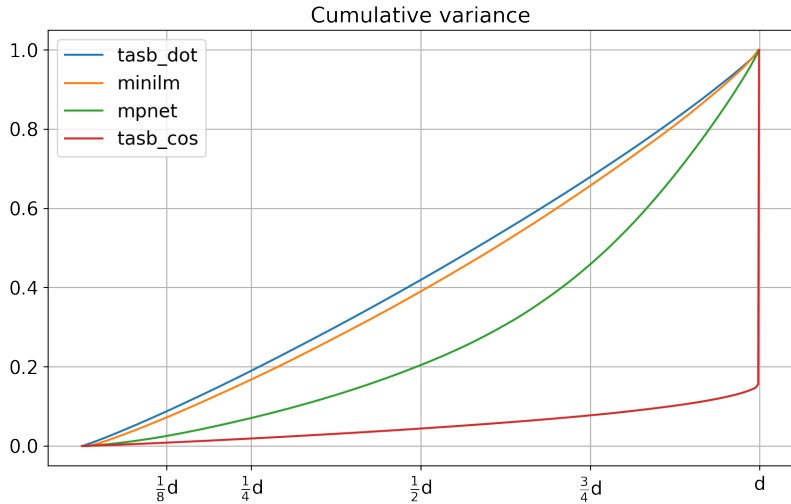

Figure 7: Cumulative variance of search transformers suggests that rogue dimensions disappear when viewed from the lens of the appropriate downstream task.

For the first claim, we have presented evidence throughout the main draft and this appendix. In Figures 3, 8 and 9 we found that vectors can be modelled as Gaussian distributions and are more isotropic than what is usually claimed to be. In Figure 2, we showed that random vectors selected from a corpus are not clumped near each other but are instead approximately orthogonal to each other in high dimensions.

For the second claim, we assess the influence of rogue dimensions by calculating the variance within that dimension across a set of vectors, as suggested by Timkey & van Schijndel (2021). Given a dataset of queries $q$ and passages $p$ we can define the random variable $cos$ for every dimension as,

$$\cos(q_i, p_i)_j \equiv \frac{q_i^j p_i^j}{|q_i||p_i|}. \tag{1}$$

Here $i$ labels the query and the passage, and $j$ refers to the coordinate and runs from 1 to $d$. We can measure the variance of these random variables for all $d$ over the entire dataset. These yield $d$ numbers, which can be used to compute the cumulative variance, i.e.,

$$F_i = \frac{\sum_{j=0}^{j=i} \text{Variance}(\cos_j)}{\sum_{j=0}^{j=d} \text{Variance}(\cos_j)} \tag{2}$$

In Figure 7, we plot the cumulative variance for `mpnet` and `minilm` and observe that search transformers do not exhibit rogue dimensions, in contrast to their `bert` counterparts, as previously reported in the literature Timkey & van Schijndel (2021). The presence of rogue dimensions would imply that the plots increase sharply as we approach $d$ on the $x$-axis, indicating that a few dimensions account for most of the variance. Note that perfectly isotropic embeddings would correspond to a straight line in Figure 7.

For the search transformer model `tasb`, we *do* observe the presence of rogue dimensions, as indicated by the red line in Figure 7. However, it's important to recognize that `tasb` has been fine-tuned with respect to Euclidean distance rather than cosine similarity. Therefore, the random variable $cos_j$ is not the appropriate metric for this context. Instead we need to define a corresponding variable,

$$\text{dot}(q_i, p_i)_j \equiv \left(q_i^j - p_i^j\right)^2. \tag{3}$$

The cumulative variance for this random variable exhibits a very smooth progression, as illustrated by the blue curve in Figure 7. We discover that rogue dimensions completely disappear when analyzed through the lens of the appropriate downstream task.

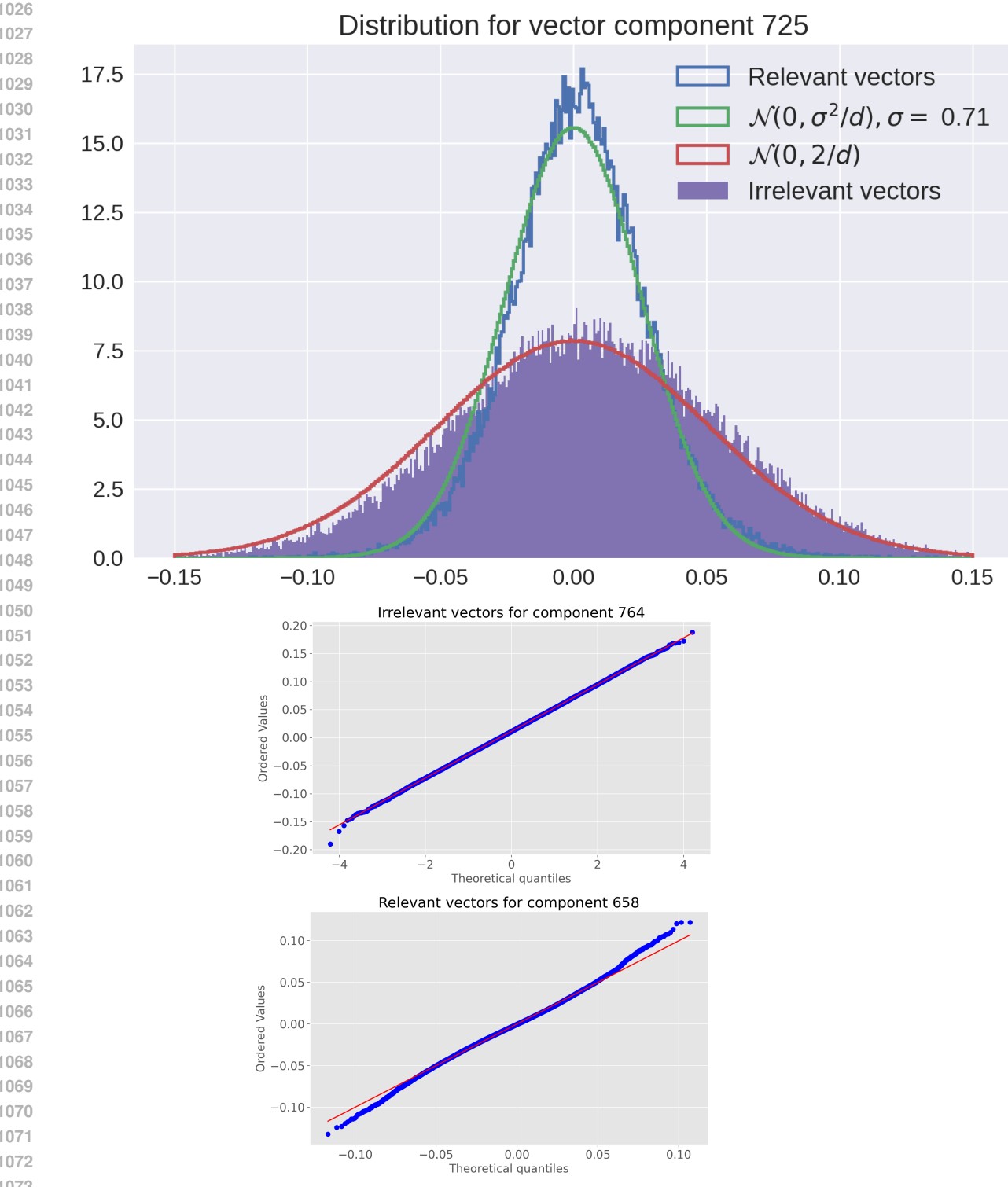

Figure 8: Histograms and $Q - Q$ plots (against normal distribution) for `mpnet` for some vector randomly selected component.

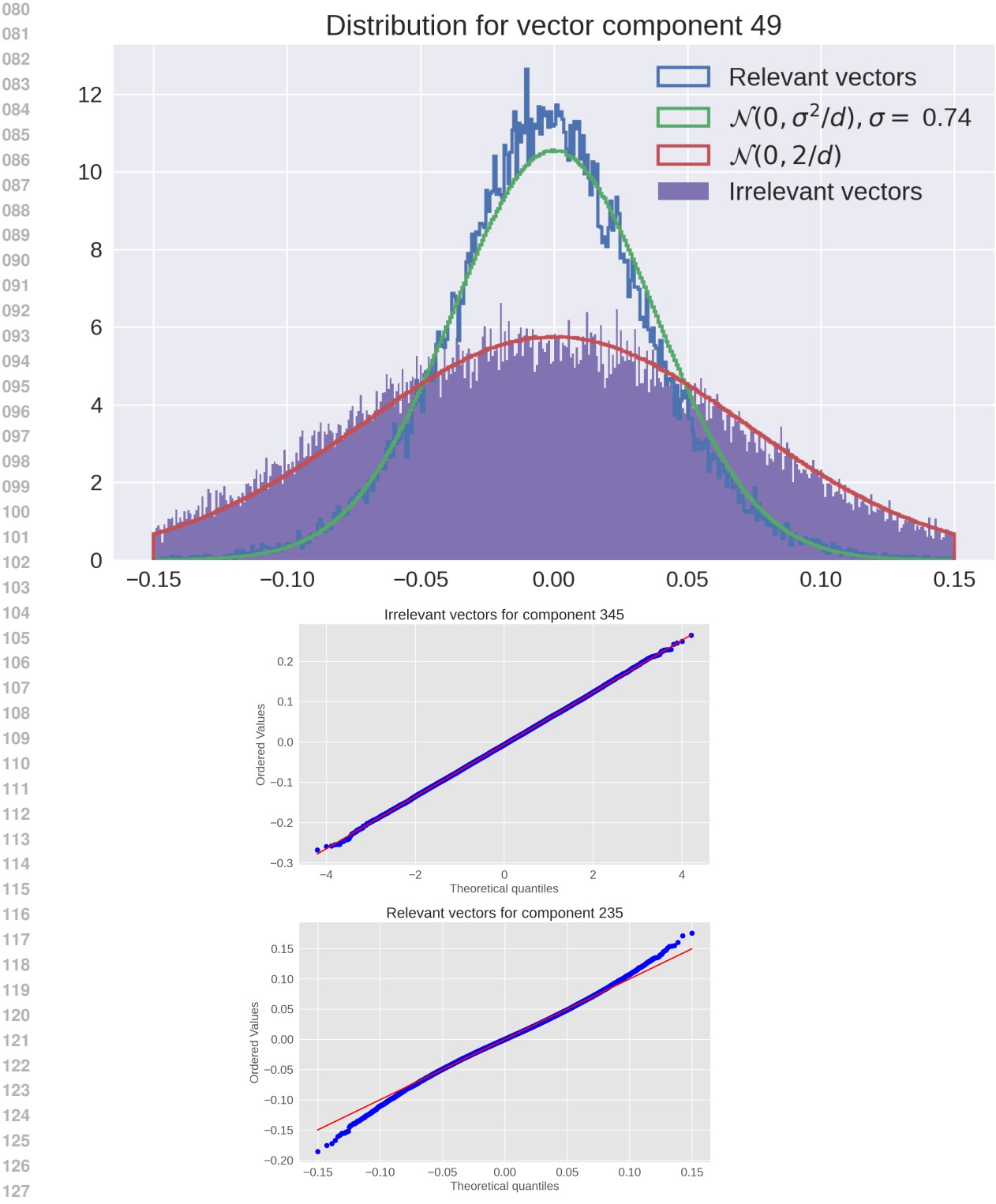

Figure 9: Histograms and $Q - Q$ plots (against normal distribution) for `minilm` for some vector randomly selected component.

