# OpenReview forum: "Filtered Semantic Search via Vector Arithmetic"
_ICLR.cc/2025/Conference — Submitted to ICLR 2025_

### Official Review · Reviewer_Kpzh · 2024-11-02

**Soundness:** 3
**Presentation:** 3
**Contribution:** 2
**Rating:** 5
**Confidence:** 3

**Summary:**

The paper is on (vector) embedding-based retrieval. Suppose we have a pre-trained (text) embedding model. Suppose we have a document corpus and assume that each document has a label (such as the \textit{color} or \textit{brand} of a product). Given a (query, label), the problem is to fetch documents which are relevant to the query and with the specified label.

The authors work in the setting where they intend to use a generic embedding model to embed the document corpus. The key proposal is to learn $d$-dimensional representations for labels (where documents are also represented in $d$-dimensional Euclidean space). The authors assume that the set of labels is small — much smaller than the corpus size. Therefore, the number of additional learned parameters is small (compared to the number of parameters in the frozen embedding model).

This is how they learn the linear probe $\nu_f$ for label f. Each document $d_i$ is assumed to have a label $y_i$. They learn a $d$-dimensional vector $\nu_f$ for each label value $f$ such that $\nu_f \cdot v_{d_i}$ is close to $\mathbf{1}\{y_i = f\}$. They minimize the squared loss $\sum_{i=1}^T (\nu_f \cdot v_{d_i} - \mathbf{1}\{y_i = f\})^2$ where $T$ is the training set size.

Now given a query and label $(q, f)$, they perform nearest-neighbor search in the document corpus for the search vector
$$
v_{q+f} + \lambda \nu_f
$$
after unit l2-normalization, where $\lambda$ is tuned by cross-validation. Note that the document embeddings are not modified, it is only the search vector that is modified, by the procedure. So crucially, this procedure can be applied for multiple label sets (filter sets) with the same document embedding corpus.

The proposed method performs significantly better than natural baselines, with no consequential increase in serving cost.


==== UPDATE AFTER THE REBUTTAL =====

There are papers like https://openreview.net/pdf?id=wLFXTAWa5V (pointed to by Reviewer BEgY) which address the same problem of filtered ANNS but with experiments on much larger datasets than those in the submission. In light of this, I believe that the authors need to demonstrate that their method scales. I am reducing my rating now, but encourage the authors to resubmit to a good conference after scaling up experiments and comparing with the methods in literature.

**Strengths:**

* The retrieval quality improvements in Table 2 are significant.

* The method is straight-forward to implement. (This is a strength.)

* Intuitively it makes sense to rank documents by the weighted sum of two scores: (i) $v_{q+f} \cdot d$: a general query document dot-product (ii) $\nu_f \cdot d$: which is trained to be close to 1 if the document is likely to have label $f$.

**Weaknesses:**

* The datasets in experiments are small (Table 1). Unfortunately, I do not have pointers to larger datasets.

**Questions:**

* In Table 2, the performance of sgpt seems to be worse than the mpnet for colors and brands; even though from the model sizes (with similar training) one would expect the opposite.

* Further, the baseline numbers are similar between the three base models. What explains this?
How is the hold-out set prepared for experiments documented in Table 3?

---

> ### Author Response · Authors · 2024-11-22
> **Response**
>
> We would like to thank the reviewer for their positive assessment as well as their interesting observations and comments.
>
> > The datasets $\ldots$ larger datasets
>
> Indeed, we are not aware of any large supervised (query, passage, filter) datasets and therefore had to create one from a dataset (Amazon ESCI) where filters are implicit. For dataset curation, we discard passages that are irrelevant to the query set or do not contain a filter. The full dataset contains over 600K documents, and we would be happy to redo experiments using it. Our initial experiments with the full dataset indicate that the recall for both the baseline and our method decreases significantly; however, the relative improvement of our method over the baseline remains consistent.
>
> > In Table 2, $\ldots$ the opposite
>
> We were puzzled by this initially as well. The comparable performance of mpnet and sgpt might be explainable through the similar accuracy of their corresponding linear probes (see Table 3, Appendix B). It is worth noting that the smallest model, minilm, has fewer output dimensions and performs the worst. Perhaps beyond a certain dimensionality (approximately the dimensionality of mpnet), additional dimensions are not necessary to represent all meaningful semantic directions, given a fixed training dataset.
>
> > Further, the baseline $\ldots$ Table 3?
>
> This is an excellent observation for which, unfortunately, we do not currently have an explanation. The train and test splits are random (with a fixed seed) across all datasets and models. We ensure that, for every dataset, each filter appears with the same frequency. This approach likely prevents our method from "cheating" by overfitting to a few popular filters at the expense of learning less common ones.

---

> > ### Comment · Reviewer_Kpzh · 2024-11-28
> > **Acknowledged response**
> >
> > I still believe the paper is a good contribution and so I keep my rating.

---

> > > ### Author Response · Authors · 2024-11-28
> > > **Thank you!**
> > >
> > > We would like to sincerely thank the reviewer for their continued support and encouragement.

---

> > > ### Author Response · Authors · 2024-12-03
> > > **Clarification Request on Score Change**
> > >
> > > Dear Reviewer,
> > >
> > > We want to thank you again for your thorough review of our paper and your positive initial feedback.
> > >
> > > We noticed that since your latest comment reaffirming your positive inclination toward the paper, your score was adjusted substantially down from 8 (Accept) to 5 (Weak Reject). Do you have any new feedback or concerns regarding the paper that we can attempt to address?

---

### Official Review · Reviewer_BEgY · 2024-11-02

**Soundness:** 2
**Presentation:** 3
**Contribution:** 2
**Rating:** 3
**Confidence:** 3

**Summary:**

This paper investigates how do efficient, vector-based nearest neighbor search given various filter criteria. The authors theoretically and empirically explore a linear probing-based approach. They show strong empirical gains using the proposed method compared to a particular baselines.

**Strengths:**

The paper explores a nuanced and import problem in nearest neighbor search, namely search for nearest neighbors meeting some filter set criteria. I believe that this is problem can have very good impact on both academic and industry research communities.

The strengths of the paper include:
* Well performing, simple, scalable approach to filtered-based nearest neighbor search
* Well presented methodological approach, clearly demonstrated through intuitive examples
* Grounding in theoretical statements

**Weaknesses:**

I am an advocate for the setting of the paper as well as the simplicity of the approach.

However, I feel that the paper falls short in a few key aspects:

* First, given the simplicity of methodology of the approach (which I generally support), I would have expected more analysis in comparison to Filtered-DiskANN and other alternative approaches. Understanding why and when to use which approach is key for having impact, especially with practitioners. I think the authors could have more clearly outlined pros & cons between their approach and Filtered-DiskANN and further shown a more complete setting of experiments to demonstrate this.
* Further, I am misguided, but I feel that the method is on the straightforward side, e.g., it is what many researchers may have wanted to try for this problem. In some ways that is a kudos for the authors for trying this, but in others, I would have expected much more explanation about the approach as it relates to classical methods like PCA, etc.
* However, my major concern about the work is that, the datasets (Table 1) are much to small by modern standards for us to get a sense of how the method scales.

Minor:
* No conclusion
* Various issues with citation parens
* There is certainly work earlier than 2009 for IR (line 116)

**Questions:**

* Can you say more about how you expect the method to scale w/ larger corpora?

---

> ### Author Response · Authors · 2024-11-21
> **Response 1**
>
> We sincerely thank the reviewer for acknowledging the simplicity of our approach and the significance of the problem statement. Please find our responses to the questions and concerns below.
>
> > Further, I am misguided, but $\ldots$ PCA, etc.
>
> We agree that the method is straightforward, especially in hindsight, but we would like to emphasize that it was our intuition that motivated us to investigate this approach, which to the best of our knowledge, has not been explored before. Our key tenet is that production-scale transformers trained via contrastive loss are incentivized to develop natural representations where directions correspond to semantic meanings or words. Our work is an attempt to invert this hypothesis and leverage it for tasks with downstream utility.
>
> We are having trouble connecting our method with the classical methods you mention. Our method neither relies on nor utilizes the covariance of the data matrix, nor does it attempt to exploit the singular values of the data. Could you please elaborate on the connection to PCA? Alternatively, while it is always possible to find the principal components of a data matrix, the existence of directions corresponding to semantic meaning is not a property that can be guaranteed a priori for an arbitrary data distribution.
>
> > However, my major concern $\ldots$ method scales
>
> We completely agree that the datasets are much smaller than today’s standards. We could certainly expand the corpus by including additional documents without applying any filters. For instance, the Amazon ESCI dataset contains more than 600K documents. In our experiments, we found that using the entire 600K dataset significantly reduced recall for both the baseline and our method. However, it is important to note that the relative improvement of our method over the baseline remained nontrivial. Please let us know if you would like us to report on our experiments with the full dataset; we would be happy to do so.
>
> > There is certainly $\ldots$ IR
>
> We apologize for the confusion—our intention was to cite a paper that reviews BM25. We will correct this in the camera-ready version and cite the original works on probabilistic retrieval from the 80s and 90s. We will include a conclusion/limitations section and address the issues with \citet vs. \citep.

---

> > ### Author Response · Authors · 2024-11-29
> > **Request for response**
> >
> > Dear Reviewer,
> >
> > We hope you are well!
> >
> > Could you please let us know if our questions and comments have addressed your concerns? We would also appreciate any techniques you have in mind regarding PCA, as mentioned earlier, and would be happy to implement them.
> >
> > We genuinely believe that our method is a meaningful contribution to the community, combining simplicity, solid experiments, and theory, while utilizing interpretability research for practical applications. We conducted extensive analyses in the appendices and experiments, built a useful evaluation dataset, and demonstrated a non-trivial improvement over a well-motivated baseline with almost no additional cost or latency.
> >
> > We hope that, in light of our rebuttal, you consider increasing your score.

---

> > ### Comment · Reviewer_BEgY · 2024-11-29
> >
> > My sincere apologies for the delay in my response.
> >
> > Thank you for your response to my review. I appreciate your comments and clarifications. However, much of my initial concerns remain after your response.
> >
> > re: scale
> >
> > Indeed, as I expressed in the initial review, I feel that for a paper about nearest neighbor search datasets up to 60K points in the index is too small. Hearing about the extension to 600K is better, but still 600K is pretty small compared to other related work, such as [Filtered-DiskANN](https://harsha-simhadri.org/pubs/Filtered-DiskANN23.pdf) which has 3.3M points, papers from ICML 2024 have [100M points](https://proceedings.mlr.press/v235/lu24l.html) and other work has scaled to billions of points for some time (e.g., [I think starting from this 2017 blog post](https://engineering.fb.com/2017/03/29/data-infrastructure/faiss-a-library-for-efficient-similarity-search/)) and now on [a single gpu](https://arxiv.org/abs/2401.11324).
> >
> > This was listed as my most major concern in my initial review and remains as such.
> >
> > As for your question about including the results, yes please do include the results.
> >
> > re: novelty
> >
> > I think the problem and method are interesting, but I feel the presentation of the method needs to make the contributions and distinctions of this simple method stand out more. E.g., it should be very straightforward for reader to understand and make decisions about when to use this method vs Filtered DiskANN etc?
> >
> > Re: related work
> >
> > I meant I would have expected a presentation that feels more complete. To provide such an example, similar kinds of methods, such as for concept erasure such as [this paper](https://proceedings.mlr.press/v162/ravfogel22a/ravfogel22a.pdf) show such connections.
> >
> > As for PCA, yes, of course, I understand what you are saying, I just meant that since you are describing these components corresponding to attributes of your data, PCA and similar methods are likely on many a readers mind (supervised, unsupervised, otherwise).
> >
> > Other possibly related work:
> > https://openreview.net/pdf?id=wLFXTAWa5V
> >
> > Thank you again for your response. My apologies for my delay in replying. I would also emphasize that I do believe the paper presents interesting ideas and settings. However, I feel the current version of the paper requires further revision before publication.

---

### Official Review · Reviewer_i6SX · 2024-11-03

**Soundness:** 2
**Presentation:** 2
**Contribution:** 2
**Rating:** 3
**Confidence:** 4

**Summary:**

The authors introduce the problem of retrieving items for queries with preference filters. They propose a simple yet effective solution of representing the resultant filtered query representation as weighted sum of the corresponding vectors. Additionally, they provide a theoretical analysis of the performance guarantees of their method under specific distributions of relevant and non-relevant items.

**Strengths:**

- Faceted search is an interesting and important problem, and the novelty of this work lies in its approach to addressing it.
- The authors present a straightforward solution and set up an experimental framework to test their method's effectiveness in solving the filtered search problem.

**Weaknesses:**

- The evaluation domains are quite limited, and some of the curated data is synthetic in nature.
- Using vector summation to produce a composite representation is a common approach. Given this, I feel the paper doesn’t present a novel contribution in representation learning. However, with its unique problem formulation and with additional analysis, along with broader testing across domains, it could still be a fit for data mining and information retrieval conferences.
- If the assumptions regarding the relative distribution of documents and query representations do not hold, the method may fail to perform effectively. Given the existing literature on the anisotropic behavior in embeddings, I am cautious about overlooking this concern.
- The theoretical claims appear to be self-evident. The authors assume that relevant and irrelevant documents follow specific distributions, then design the representation as the distribution mean with a separability threshold based on variance. Naturally, this setup ensures separability; thus, the analysis seems to add little value. It would be more compelling if the authors showed that the method performs within an ϵ-optimal range even when data distributions deviate from these assumptions.

**Questions:**

The weakness themselves can be considered as questions.

**Additional Questions and Suggestions:**

- Of the following three contributions, which do the authors consider most central to their work?
    - Identifying the problem of filtered search and providing a benchmark for it
    - Proposing a non-trivial method to solve the filtered search problem
    - Offering theoretical guarantees for the proposed method

    The paper touches on aspects of each of these contributions, but none are explored in depth. I suggest enhancing one or more of these areas and restructuring the paper with a clearer, more focused motivation. While focusing on all of them is a good idea, but the paper reads very incomplete without going into depth in some of them.

- Do the authors believe similar ideas could extend to other domains? The current filters seem somewhat synthetic—additional experiments on real-world data would strengthen the work.
- Could you test alternative, simple baselines?
    - For instance, multiply the scores of the query and filter for each document, such that

        p(q+f,d)=p(q,d)×p(f,d).

    - Another approach could involve converting  q+f into natural language queries using an LLM, then feeding the output text to the encoder to generate the query + filter representation

---

> ### Author Response · Authors · 2024-11-21
> **Response 1**
>
> We genuinely thank the reviewer for a very thorough and well thought review. Below we aim to address the individual questions/comments,
>
> > The evaluation domains $\ldots$ in nature.
>
> We acknowledge that the evaluation domains are limited but would like to emphasize the scarcity of labeled (query, passage, filter) datasets. Notably, one of our three datasets consists of synthetically created filters. While these filters are synthetic, they precisely measure the performance of our technique for the given task—retrieving documents that satisfy strict keyword criteria. We would be delighted to benchmark our method on additional datasets; if you are aware of any, please let us know.
>
> Additionally, we attempted to benchmark our method using image datasets such as MS COCO [1], ImageNet [2], and the Caltech Birds [3] dataset. In this context, the image serves as the document to be retrieved (e.g., an image of a child playing with a dog), the query is expressed in plain English (e.g., “kids playing with pets”), and the filter could be an attribute such as the color or breed of the dog. Unfortunately, creating high-quality (query, passage, filter) triples from these datasets requires significant manual effort, and we were unable to automate this process effectively. We do have anecdotal results (e.g., the example shown in Figure 1 featuring the “red sweater”) and would be happy to include an additional appendix detailing these results.
>
> > If the assumption $\ldots$ this concern
>
> Our theoretical bounds rely on the modeling assumptions of the document/query representations. Under these assumptions, we are able to derive results that are significantly stronger than those observed in our experiments. However, the practical effectiveness of our method arises from the presence of directions in the embedding space that faithfully represent specific semantic concepts or words. In this regard, our approach is related to the superposition hypothesis [4,5]. The existence of these directions in large neural networks remains an open question and has only been studied in simplified toy models until recently [6]. To the best of our knowledge, no prior work has investigated this hypothesis in the context of production-scale search transformers or applied it to tasks of downstream importance.
>
> The presence of anisotropy does not necessarily inhibit our method’s effectiveness (although it complicates theoretical analysis); it may simply make identifying the linear probe more challenging. As discussed near Figure 2 and in Appendix E, concerns about anisotropic transformer embeddings may be somewhat overstated. For unit-norm models, we observe significantly reduced anisotropy, which is indeed the information theoretically favored solution.
>
> In Appendix D.4, we present experiments using a model with non-normalized embeddings. We find that the results are slightly worse compared to those obtained with unit-norm models. Unfortunately, we have not yet thoroughly investigated this discrepancy. A plausible explanation could be the increased difficulty of identifying a high-fidelity linear probe in this setting. We will highlight this observation in the camera-ready version.
>
> > The theoretical claims $\ldots$ these assumptions
>
> Our model does not inherently assume any separability, and in many instantiations the two distributions are largely overlapping. For example, for small values of $\sigma$ and $\gamma$, the sphere of relevant vectors may be partially or entirely contained within that of the irrelevant ones. What we argue is that, supposing that a filter direction does actually exist, and that relevant documents satisfying the filter do cluster around some $v_q + \gamma \nu_f$ (with $\gamma$ unknown), then moving the query vector $v_q$ by any amount $\lambda$ in the direction of $\nu_f$ will eliminate irrelevant vectors from the list of k-NNs at a rate depending on $|\lambda - \gamma|$ and exponential in $d$ (lines 410 and 414).
>
> In particular, while we still see an advantage when $\lambda$ is far from $\gamma$, choosing a value of $\lambda$ close to $\gamma$ increases the rate at which we converge to a filter-satisfying set of nearest neighbors. This can be made more explicit by rewriting the lemma to include this dependence. For example, by fixing $\lambda$ = 0.4 (the midpoint of the range for the unknown parameter γ, thereby bounding $|\lambda  - \gamma|$ ≤ 0.4) the value 0.943 can be improved to 0.887, which is more than can be said for a fixed $\lambda = 0$, thereby justifying the addition of the scaled filter probe. We will incorporate this calculation into the paper.

---

> ### Author Response · Authors · 2024-11-21
> **Response 2**
>
> > Of the following three contributions $\ldots$ work?
>
> This is a fair question, and we sincerely appreciate it. We believe that our main contribution lies in proposing a non-trivial (yet simple) method for solving the filtered search problem. The rest—the theoretical framework, the empirical findings on search embeddings, and the benchmark—are efforts to bring this idea to fruition.
>
> Our approach hinges on the hypothesis of a correspondence between directions in the embedding space and semantic meanings. To the best of our knowledge, this hypothesis has previously been explored only in toy models and with methods requiring substantial manual supervision. Instead, we assume this hypothesis as a given and leverage it for a meaningful downstream task. To this end, we developed a realistic benchmark and obtained non-trivial results. In the process, we made empirical discoveries (e.g., the discussion near Figure 2 on page 4 and Appendix E), derived theoretical bounds, and tested the assumptions underlying our analysis.
>
> In our humble opinion, our simple yet effective method is a valuable contribution to the broader community, offering simplicity, empirical rigor, theoretical insight, and a novel perspective on utilizing interpretability research for downstream utility.
>
> > Do the authors believe similar $\ldots$ the work?
>
> As mentioned above, we are confident that our approach can be extended to other domains, particularly for embeddings derived from unit-norm search transformers. We would be happy to include an additional appendix detailing our experiments with MS Coco. If you have any other datasets in mind, please do let us know.
>
> > Could you test $\ldots$ baselines?
>
> We are not sure if we fully understand the suggestion to multiply the scores. Since kNN search produces a distribution of distances, we are unaware of a canonical way to convert these into probability distributions. Could you please elaborate?
>
> The LLM-based approach for rewriting the query is indeed an interesting idea, but we believe it is orthogonal to our method—that is, our method could be used alongside the LLM-rewritten query. However, we are also unsure how to effectively control the drift of the LLM-rewritten query from the original user query. For instance, an LLM could rewrite the query "pajamas red" as "pajamas Christmas red maroon holiday dark-red," which might yield more results satisfying the filter "red" but diverge from the user’s intended meaning of "pajamas."
>
> Finally, we would like to conclude by reiterating our gratitude to the reviewer for their insightful questions and concerns. Their feedback has greatly helped us refine our understanding and think about our work more clearly.
>
> [1] https://cocodataset.org/#home
>
> [2] https://www.image-net.org/
>
> [3] https://authors.library.caltech.edu/records/cvm3y-5hh21
>
> [4] https://transformer-circuits.pub/2022/toy_model/index.html
>
> [5] https://transformer-circuits.pub/2023/monosemantic-features/index.html
>
> [6] https://transformer-circuits.pub/2024/scaling-monosemanticity/index.html

---

> > ### Comment · Reviewer_i6SX · 2024-11-25
> > **Reviewer Response**
> >
> > 1. For normalized embeddings, distance= 2 - 2 * cosine_similarity. Calculate the cosine similarity and then pass it through a sigmoid to get the probability (you need to be careful with the temperature.)
> >
> > 2. I would suggest the authors to extend to multiple domains with thorough experiments.
> >
> > 3. Include more embedding models, very large models. With an analysis of measurement of anisotropy and relative gains over the model.
> >
> > 4. I suspect with good enough in context examples, GPT won't generate something like "pajamas Christmas red maroon holiday dark-red" .
> >
> > '''
> > I would give you object names and then an attribute corresponding to the class of object. You need to generate a natural language query that a human would search to get the object that satisfies the constraint attribute.
> >
> > Object: Pajamas Attribute: Red.
> >
> >
> > Answer: "Find me red pajamas."
> >
> >
> > The proposed solution is the simplest, such combinations are common from the days of Word2vec or GloVe. I am not discouraging the solution, but it would require significant demonstration of usefulness for it to be accepted as a novel work.

---

> > > ### Author Response · Authors · 2024-11-29
> > > **Response and request to increase the score**
> > >
> > > We would like to thank the reviewer for their continued engagement. We really appreciate it. Please find our comments below,
> > >
> > >
> > > > For normalized $\ldots$
> > >
> > >
> > > Since cosine is always between -1 and 1 and sigmoid is roughly linear in this range, this is similar to multiplying the cosine scores. However, we are uncertain if direct multiplication is the most suitable method, since we want the filtration criteria to be strictly satisfied. Perhaps multiplication with each term raised to a different power might be more relevant, but identifying the right approach (as with the temperature scaling the reviewer insightfully mentioned) would require extensive experimentation. Additionally, this method would require twice the number of queries to approximate kNN: one for the query vector and one for the filter vector.
> > >
> > > > I would suggest $\ldots$
> > >
> > > We sincerely agree with the spirit of your comment but would like to emphasize the scarcity of labeled $(q, p, f)$ datasets, as also highlighted by Reviewer Kpzh. In our work, we have attempted to address this by creating a high-quality benchmark through careful manipulation of the Amazon ESCI dataset. We would greatly appreciate any suggestions you might have.
> > >
> > >
> > > > Include more embedding models, very large models. $\ldots$
> > >
> > >
> > > We believe that our work spans a well-motivated range of model sizes (from 22M to 1.3B), striking a balance between industry requirements and SOTA accuracy. Embedding models are typically significantly smaller than decoder models, with SOTA models typically ranging from 1B to 5B parameters — indeed on the MTEB benchmark the difference between embedding models with 1024 dimensions (~ 1B models) and 4096 (~ 5B models) is less than tenth of a percent.
> > >
> > > Regarding the analysis of anisotropy, could you please clarify your suggestion? Are you proposing repeating the analysis in Appendix E for additional models, or do you have a specific technique in mind that you would like us to apply for the models we use?
> > >
> > >
> > > > I suspect with good enough $\ldots$
> > >
> > >
> > > As mentioned above, this is an idea that can be utilized in tandem with our technique. The combination used for our experiments $v_{q+f} + \lambda\nu_f$ automatically allows for such a method. The new query simply becomes $v_{q_{LLM}} + \lambda\nu_f$.
> > >
> > > As an aside, we believe that “pajamas Christmas red maroon holiday dark-red” is  (in this context) a better rewritten query than “Find me red pajamas,” since embedding models are more likely to retrieve pajamas that are red in color with the former. This is because, in our view, embedding models (unlike decoder LLMs) trained with contrastive loss are better suited to prescriptive queries than descriptive ones.
> > >
> > > **Summary**
> > >
> > > At the risk of repetition, we would like to reiterate our response to the reviewer’s thoughtful and well-motivated question:
> > >
> > > > Of the following three contributions, which do the authors consider most central to their work?
> > >
> > >
> > > We are not claiming that our method of combination is novel. As the reviewer rightly pointed out (and as cited in our work), this approach has been known since the days of Word2Vec.
> > >
> > > In our humble opinion, our method represents a meaningful contribution to the broader community by offering a balance of simplicity, empirical rigor, theoretical insight, and a novel way of leveraging interpretability research for downstream applications. To realize this idea, we conducted a rigorous analysis grounded in theory and empirical findings on search transformers, created an appropriate evaluation dataset, and achieved a non-trivial boost over a well-motivated baseline with almost no additional cost or latency. The novelty is not in the concept of combining vectors, but in the specific methodology borne out of geometric insights about the space they reside in.
> > >
> > > We're also interested to hear if your concerns about the "theoretical claims being self-evident" have been sufficiently addressed. We would like to reiterate that our model does not inherently assume any separability, and in many instantiations the two distributions are actually overlapping!
> > >
> > > Finally, we hope you find our approach promising for the wider community and consider raising your score.

---

### Meta-Review · Area_Chair_UvE6 · 2024-12-21

**Metareview:**

The paper considers the problem of semantic search with certain filtering criteria. It proposes to identify a vector corresponding to each filter, and use this to augment the query vector to achieve filtering. A theoretical analysis is presented to quantify the probability of such an approach yielding irrelevant results.

Reviewers generally found the paper to consider an interesting problem, but considered the technical novelty to be somewhat modest and narrow in scope. The work is primarily empirical in nature, but the presented results are on small-scale datasets. Further, a number of baselines were suggested by reviewers, which would strengthen the comparisons and help identify the value of individual components of the proposal. Finally, the presentation was adjudged to have room for improvement.

Overall, the paper appears best suited to revision and submission to some future conference.

**Additional Comments On Reviewer Discussion:**

Reviewers engaged with the author response, but generally found their significant concerns to be left open (e.g., scale of experiments, comparison to baselines). Following discussion, one reviewer who was initially positive agreed with the other reviewers about scope for improvement, and amended their score to be slightly below the acceptance threshold.

---

### Decision · Program_Chairs · 2025-01-22

Reject